# THE EFFICIENCY MISNOMER

**Mostafa Dehghani**[*], **Anurag Arnab**[*], **Lucas Beyer**[*], **Ashish Vaswani, Yi Tay**[*]
Google Research
`{dehghani, aarnab, lbeyer, avaswani, yitay}@google.com`

## ABSTRACT

Model efficiency is a critical aspect of developing and deploying machine learning models. Inference time and latency directly affect the user experience, and some applications have hard requirements. In addition to inference costs, model training also have direct financial and environmental impacts. Although there are numerous well-established metrics (cost indicators) for measuring model efficiency, researchers and practitioners often assume that these metrics are correlated with each other and report only few of them. In this paper, we thoroughly discuss common cost indicators, their advantages and disadvantages, and how they can contradict each other. We demonstrate how incomplete reporting of cost indicators can lead to partial conclusions and a blurred or incomplete picture of the practical considerations of different models. We further present suggestions to improve reporting of efficiency metrics.

## 1 INTRODUCTION

Aside from model quality, the efficiency (Menghani, 2021) of a model is often an important aspect to consider and is commonly used to measure the relative utility of different methods. After all, training time spent on accelerators is directly linked to financial costs and environmental impact. Meanwhile, the speed of a model may be directly linked to user experience. To this end, there have been well-established ways in the literature to assess and report the efficiency of a model such as number of trainable parameters, number of floating-point operations (FLOPs), and speed/throughput.

While it is commonly assumed that these cost indicators are correlated (e.g., a lower number of parameters would translate to a higher throughput) we show that this might not necessarily be the case. Therefore, incomplete reporting across the spectrum of cost indicators may lead to an incomplete picture of the metrics, advantages and drawbacks of the proposed method. To this end, we show that it may also be possible to, perhaps unknowingly, misrepresent a model's efficiency by only reporting favorable cost indicators. Moreover, the choice of cost indicators may also result in unfair, incomplete, or partial conclusions pertaining to model comparisons. We refer to this phenomenon as the '*efficiency misnomer*'.

The overall gist of the *efficiency misnomer* is that no single cost indicator is sufficient. Incomplete reporting (e.g., showing only FLOPs, or the number of trainable parameters) as a measure of efficiency can be misleading. For example, a model with low FLOPs may not actually be fast, given that FLOPs does not take into account information such as degree of parallelism (e.g., depth, recurrence) or hardware-related details like the cost of a memory access. Despite this, FLOPs has been used as the most common cost indicator in many research papers, especially in the recent computer vision literature, to quantify model efficiency (Szegedy et al., 2015; He et al., 2016; Tan and Le, 2019; Feichtenhofer et al., 2019; Fan et al., 2021).

Likewise, the number of trainable parameters (size of the model) despite being commonly used as the de-facto cost indicator in the NLP community (Devlin et al., 2018; Liu et al., 2019; Lan et al., 2019) and previously the vision community (Krizhevsky et al., 2012; Simonyan and Zisserman, 2015; Huang et al., 2017; Tan and Le, 2019), can also be misleading when used as a standalone measure of efficiency. Intuitively, a model can have very few trainable parameters and still be very slow, for instance when the parameters are shared among many computational steps (Lan et al., 2019; Dehghani et al., 2018). While the number of trainable parameters can often be insightful to decide if a model fits in memory, it is unlikely to be useful as a standalone cost indicator. That said, it is still common practice to *parameter-match* models to make *'fair'* comparisons (Mehta et al., 2020; Lee-Thorp et al., 2021; Tay et al., 2020a; Xue et al., 2021; Wightman et al., 2021), even if one model is in reality slower or faster than another.

---

[*]Equal contribution.

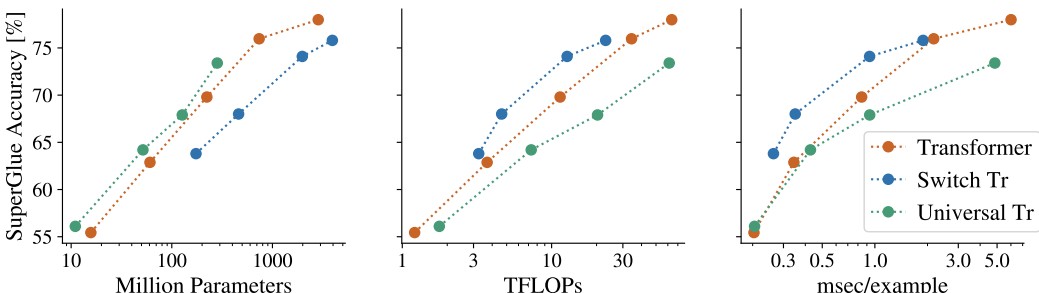

Figure 1: Comparison of standard Transformers, Universal Transformers and Switch Transformers in terms of three common cost metrics: number of parameters, FLOPs, and throughput. Relative ranking between models is reversed between the two cost indicators. Experiments and the computation of cost metrics were done with Mesh Tensorflow (Shazeer et al., 2018), using 64 TPU-V3.

Meanwhile, using throughput/speed as the primary indicator of efficiency can also be problematic - since this tightly couples implementation details, hardware optimizations, and infrastructure details (e.g., input pipeline latency) into the picture. Hence, this might not present an apples-to-apples comparison of certain methods or worse, across different infrastructures or hardware.

Given that the landscape of research on model architectures is diverse, the relationship between cost indicators may strongly deviate from the norm, and learning how to fairly compare models within the context of efficiency-based cost indicators is crucial. For instance, there seems to be a rising trend towards sparse models (Fedus et al., 2021; Riquelme et al., 2021) which usually have an incredibly large number of trainable parameters but maintain the FLOPs and speed of dense models. On the contrary, there are models that are considered lightweight due to their small number of trainable parameters (Lan et al., 2019; Dehghani et al., 2018) but, in actual practice, consume similar amounts of compute. Figure 1 shows an example of how the scaling behavior of a model with respect to parameter count can look favorable, while taking the FLOPs or throughput as the cost indicator, different model scales much better. This example shows how looking at one metric can be deceptive and cost a lot of time and resources, e.g. by choosing the wrong candidate for scaling up.

Besides the fact that different cost indicators capture different aspects, they can be chosen to reflect either the cost of training or the cost at inference time. Both training and inference costs can be crucial, depending on the context. Also, a single cost indicator can favor a model over another during inference, but not training (or vice versa). For instance, a model that shares parameters in depth is particularly memory-efficient during inference, but during training, the size of activation that need to be kept for the backward pass is just as large as for a similar model with no parameter sharing.

The overarching conundrum here is that first of all, no single cost indicator captures a holistic view that is universally useful to all practitioners or researchers. We show that the trade-offs between cost indicators fall far from the standard assumptions and can be non-trivial to navigate. Moreover, we argue that cost indicators that one cares about strongly depend on the applications and setup in which the models are supposed to be used. For example, for an embedded application, inference speed is paramount, while for deployed recommendation systems, training cost can be extremely important as models are constantly being retrained.

The overall contributions of this paper are as follows: We call out the intrinsic difficulty of measuring model efficiency within the context of deep neural networks. We review the most common cost indicators and present the advantages and disadvantages of each and discuss why they might be insufficient as a standalone metric. While obvious, we show examples of how model efficiency might be misrepresented by incomplete reporting of these cost indicators. We characterize this problem, coin the term *'efficiency misnomer'*, and show that it is more prevalent than imagined.

We present experiments where comparing model efficiency strongly depends on the choice of cost indicator, like scenarios where there is parameter sharing, sparsity, or parallelizable operations in the model. Moreover, we briefly review some of the current common practices in the literature and discuss how existing work report comparisons of different models and analyze the efficiency of algorithms. Along with the discussion and analyses, we provide some concrete suggestions and recommendations that we believe would help researchers and practitioners draw more accurate conclusions about the efficiency of different models.

## 2 A PRIMER ON COST INDICATORS

One of the main considerations in designing neural network architectures is quality-cost trade-off (Paleyes et al., 2020). In almost all cases, the more computational budget is given to a method, the better the quality of its outcome will be. To account for such a trade-off, several cost indicators are used in the literature of machine learning and its applications to showcase the efficiency of different models. These indicators take different points of view to the computational costs.

**FLOPs:** A widely used metric as the proxy for the computational cost of a model is the number of floating-point multiplication-and-addition operations (Johnson, 2018; Kim et al., 2021; Arnab et al., 2021; Tay et al., 2021a;b; Narayanan et al., 2021; Liu et al., 2021). Alternative to FLOPs, the number of multiply-accumulate (MAC[1]) as a single unit of operation is also used in the literature (Johnson, 2018). Reported FLOPs are usually calculated using theoretical values. Note that theoretical FLOPs ignores practical factors, like which parts of the model can be parallelized.

**Number of Parameters:** Number of trainable parameters is also used as an indirect indicator of computational complexity as well as memory usage (during inference) (Kim et al., 2021; Arnab et al., 2021; Tan and Le, 2019; Liu et al., 2021; Guo et al., 2020; Mahabadi et al., 2021a;b; Houlsby et al., 2019). Many research works that study the scaling law (Kaplan et al., 2020; Hernandez et al., 2021; Tay et al., 2021b), especially in the NLP domain, use the number of parameters as the primary cost indicator (Devlin et al., 2018; Raffel et al., 2019; Liu et al., 2019; Xue et al., 2021).

**Speed:** Speed is one the most informative indicator for comparing the efficiency of different models (So et al., 2021; Dosovitskiy et al., 2020; Arnab et al., 2021; Tay et al., 2021b; Kim et al., 2021; He et al., 2021a; Narayanan et al., 2021; Lagunas et al., 2021; Liu et al., 2021; Tay et al., 2020c;b). In some setups, when measuring speed, the cost of "pipeline" is also taken into account which better reflects the efficiency in a real-world scenario. Note that speed strongly depends on hardware and implementation, so keeping the hardware fixed or normalizing based on the amount of resources used is the key for a fair comparison. Speed is often reported in various forms:

- *Throughput* refers to the number of examples (or tokens) that are processed within a specific period of time, e.g., "examples (or tokens) per second".
- *Latency* usually refers to the inference time (forward pass) of the model given an example or batch of examples, and is usually presented as "seconds per forward pass". The main point about latency is that compared to throughput, it ignores parallelism introduced by batching examples. As an example, when processing a batch of 100 examples in 1 second, throughput is 100 examples per second, while latency is 1 second. Thus, latency is an important factor for real-time systems that require user input.
- *Wall-clock time/runtime* measures the time spent to process a fixed set of examples by the model. This is often used to measure the training cost, e.g., total training time up to convergence.
- *Pipeline bubble* is the time that computing devices are idle at the start and end of every batch (Narayanan et al., 2021), which indirectly measures the speed of the non-pipeline parts of the process.
- *Memory Access Cost (MAC)* corresponds to the number of memory accesses. It typically makes up a large portion of runtime and is the actual bottleneck when running on modern platforms with strong computational power such as GPUs and TPUs (Ma et al., 2018).

The cost indicators we discussed above present different perspectives on efficiency. However, some of these cost indicators may depend on factors that are not inherent to the design of the model, but on the hardware, the model runs on (e.g., CPU, GPU, or TPU), the framework that the model is implemented in (e.g., JAX, PyTorch, or TensorFlow), or even programming skill. These confounding factors add up to the difficulty of comparisons. For instance, theoretical FLOPs provides a hardware-independent comparison, however, it does not necessarily translate to the speed of a model as it does not capture the sequential dependencies of operations and memory accesses in the model. On the other hand, throughput and peak memory usage, which could better reflect the model's efficiency in a real-world scenario, strongly depend on the hardware and the implementation. Software support can also be a limiting factor in achieving the best possible hardware performance for a model. In (Barham and Isard, 2019), the authors make an excellent case for improving the programmability of software stack for modern accelerators to enable a wider class of models.

---

[1]MAC may also refer to the memory access cost (Ma et al., 2018)

In the rest of the paper, we mainly focus on the number of parameters, FLOPs and speed since they capture most common use cases and are most commonly encountered in the literature. Appendix B presents other cost indicators that could be important depending on the use case.

## 2.1 TRAINING OR INFERENCE COST?

When talking about costs, we can disentangle the cost of training and the cost of inference. Based on the estimate from NVIDIA (Leopold, 2019) and Amazon (Barr, 2019) as major cloud service providers, 80–90% of the ML workload is inference processing. Thus, more often, the cost of a model during inference is taken as the real cost with the argument that the amortized per-usage cost of training can be really small compared to the inference cost when a model is deployed to be used by many users. However, with the trend of improving the performance of models by scaling up their computational budget and/or training data, as well as the fast progress and frequency of the emergence of new models, the training cost can be seen as a relevant concern. Moreover, in some cases, we have to frequently retrain models due to, for instance, privacy requirements, or where new data is being continuously generated like in many recommender systems. The importance of inference efficiency is already clear and here, we will discuss more the importance of training efficiency as well as potential issues with reporting training costs.

The are several arguments in favor of the importance of reporting training costs and the need for models that train efficiently. For instance, from a research and development point of view, when a class of models is efficient during training, it is more likely to see improvements in their performance, due to ease of iterating on ideas around them. Moreover, when a model shows merit in terms of performance, usually some posthoc modifications can be applied to improve its inference efficiency.

Besides, the memory requirements of different models can be wildly different during training, although their inference memory consumption is comparable. For instance, different optimizers use different amounts of memory on the device, for instance, SGD-Momentum (Qian, 1999) vs SAM (Foret et al., 2020). If the success of a model is strongly tied to using an optimizer with a high memory cost, it can become a bottleneck during training when we plan to scale that model up compared to the model that uses a more memory-efficient optimizer.

Another example is a model that has a high degree of parameter sharing, which could be extremely efficient in terms of inference memory usage, but during training, the size of activation we need to keep for the backward pass is as big as a similar model with no parameter sharing. Activation and their statistics form a big portion of memory usage compared to parameter size, decreasing the "number of parameters" by parameter sharing may not lead to any significant decrease in the memory usage during training (Dehghani et al., 2018).

**Gaming with training time** Whilst "training time" can be a great cost indicator, it is also prone to Goodhart's Law: When used as the main metric, it can and will be gamed and lose its meaning. First of all, it is difficult to compare *architectures* with respect to training time, since training time includes the whole "recipe" and different ingredients may be better fits for different architectures. For instance, MobileNet and EfficientNets almost exclusively work well with the RMSProp optimizer (Pham, 2021).

Second, because the full training recipe is inevitably involved, the only meaningful claim to be made is achieving higher accuracy with smaller total training cost; a good example of this is Table 1 in Wightman et al. (2021). The training cost may be in terms of any of the indicators described above. However, counting training cost in terms of steps can be problematic as step count in itself is not a meaningful metric, and can be stretched arbitrarily with optimizers introducing multi-step lookahead schemes (Zhang et al., 2019).

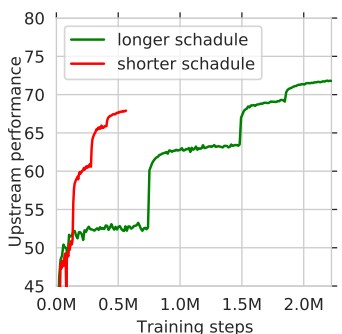

Figure 2: The learning progress of a ResNet-101 × 3 on JFT-300M with short and long schedules, obtained from (Kolesnikov et al., 2020). Decaying the learning rate too early leads to higher performance in lower steps, but the final performance is significantly worse.

Conversely, it may be tempting to claim that a method A performs *almost as well* as another method B with *dramatically reduced training cost*. Such a claim is not valid, as method A may not have been optimized for training cost, and may well just be re-tuned with that in mind and outperform method B. For example, training hyper-parameters such as learning rate and weight decay, can be tuned such

that they achieve good quality quickly, but then plateau to lower points than in "slower" settings that eventually reach higher quality. This is illustrated by Figure 2, obtained from Kolesnikov et al. (2020), where the "long" training schedule (which is not optimized for training cost) for ResNet-101x3 achieves the best performance, but the "short" schedule converges significantly faster to a lower final accuracy. Another example, in the context of reinforcement learning, is the Rainbow baseline of Kaiser et al. (2019) which was subsequently re-tuned in van Hasselt et al. (2019), and shown to benefit from significantly longer training schedules too.

Some works, like the MLPerf benchmark (Mattson et al., 2019), aim at decreasing the training cost required to reach a fixed quality X with a fixed model M. While this is a good way of fixing the many moving pieces of training a deep learning model, it should be noted that due to Goodhart's law, results do not mean more than "reaching quality X with model M". Specifically, if a method A reaches quality X twice as fast as a method B while both using model M, this can neither be used to imply anything about their comparison when using model N, nor to imply anything on their comparison with the target quality X+$\epsilon$: it could well be that method B gets to X+$\epsilon$ faster than A, or worse, A may never reach X+$\epsilon$. Figure 3 shows another concrete example, obtained from (Kolesnikov et al., 2020) where we see faster initial convergence of a ResNet model when trained with lower weight decay, which may trick the practitioner into selecting a sub-optimal value, while using a higher weight decay leads to a slower convergence, but a better final performance.

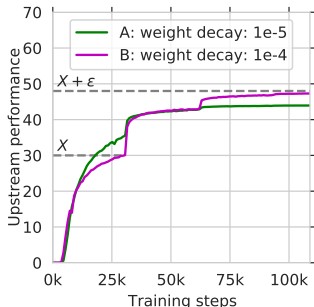

Figure 3: The learning progress of a ResNet-101 $\times$ 3 on JFT-300M with different weight decays, obtained from (Kolesnikov et al., 2020). Lower weight decay leads to acceleration of convergence, while eventually results in an under-performing final model.

A final concern is that, methods which seemingly train faster tend to be used more often and hence get optimized more over time, with the danger of getting stuck in a local minimum (Hooker, 2020; Dehghani et al., 2021b).

## 2.2 POTENTIAL DISAGREEMENT BETWEEN COST INDICATORS

In this section, we will discuss some of the cases in which there could be disagreement between some of the cost indicators we discussed before.

**Sharing parameters** When we introduce a form of parameter sharing, it is clear that compared to the same model with no parameter sharing, we end up having fewer trainable parameters, while the number of FLOPs or speed stays the same. An example is the comparison of the Universal Transformer (UT) (Dehghani et al., 2018) with vanilla Transformer (Vaswani et al., 2017) Figure 1. UT shares parameters of the model in depth, thus stays close to the frontiers of quality-number of parameters. However, when looking at the FLOPs, to maintain a similar capacity in terms of parameter count, UT requires more computation, which makes it not so efficient from the FLOPs point of view.

**Introducing Sparsity** Sparsity is becoming one of the main ways of both scaling up and down deep neural networks. One needs to distinguish between at least two broad classes of sparse neural networks: structured and unstructured.

Structured sparse models replace large, dense parts of a model by a collection of much smaller, still dense parts. Examples include variants as simple as replacing dense convolutions by grouped (Xie et al., 2017) or separable (Howard et al., 2017) ones, or as complicated as replacing large blocks by many smaller "expert" blocks and routing examples through the best suited ones only (Fedus et al., 2021; Riquelme et al., 2021). The latter, often called Mixture of Experts (MoE), allows growing the capacity in terms of parameter count, while keeping the computational cost small and constant. Figure 1 compares the quality-cost of Switch Transformer (Fedus et al., 2021), a MoE, to that of vanilla Transformer (Vaswani et al., 2017). While Switch falls short in terms of quality vs parameter count, it offers a great trade-off with respect to quality vs FLOPs and speed.

Unstructured sparse models are models where weights of dense operations are made to contain many (almost) zeros (Gale et al., 2019; Evci et al., 2020), which do not contribute to the operation's result and thus, in principle, can be skipped. This keeps the overall structure of the original model, while significantly reducing the FLOPs of the most expensive operations.

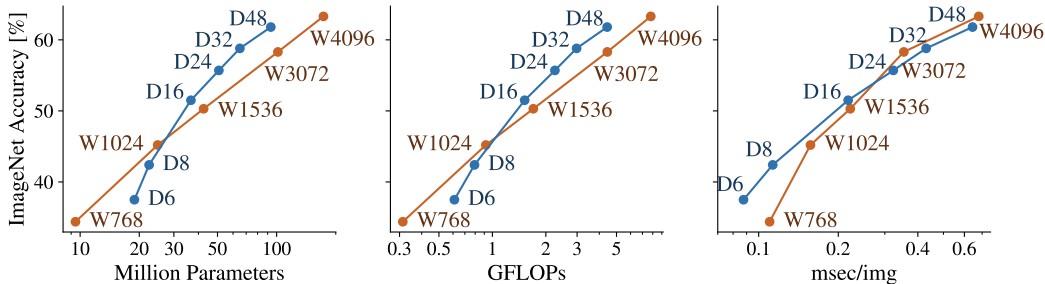

Figure 4: Comparison of scaling a small ViT in depth (D, number of encoder blocks) vs scaling it in width (W, hidden dimension). Which architecture appears "better", in terms of cost-quality trade-off, changes depending on which indicator is considered. Experiments and the computation of cost metrics were done with Scenic (Dehghani et al., 2021a), using 64 TPU-V3.

Both types of sparse models result in large reductions in theoretical FLOPs, often of several orders of magnitude. However, these do not translate to equally large speed-ups. Difficulties for structured sparse models (especially for the MoE type) include the overhead of routing, and the inability to effectively use batched operations, while for unstructured sparsity, it is not possible for the corresponding low-level operations to reach the same efficiency as their dense counterparts on current hardware, where memory access is significantly more expensive than compute (Gale et al., 2020).

**Degree of parallelism: Scaling Depth (D) vs. Scaling Width (W)** When scaling up the model size, different strategies can be used. Although these different strategies may have a similar effect in terms of parameter count, and even quality, they can have different effects on the cost in terms of FLOPs and throughput. The most common knobs for scaling up models are changing depth (number of layers) and width (hidden dimension) of models (Tay et al., 2021b). To study such an effect, we ran a set of controlled experiments with Vision Transformers (Dosovitskiy et al., 2020), where we scale the width by increasing number of heads (while maintaining the hidden dimensions of heads fixed) as well as that of the FFN, and we also scale up the depth, by only increasing the number of encoder blocks. Note that when changing depth or width of the model (see Table 2 in Appendix A for the exact configurations) all other hyper-parameters are kept fixed based on the default values given by the referenced papers. Figure 4 shows the accuracy of these models with respect to different cost indicators.

In general, we can see that considering FLOPs or number parameters as the cost indicator, increasing width is beneficial for smaller budget regions, while increasing depth gives better quality with lower cost when we scale up to higher budget regions. However, this is not necessarily the case when considering speed (msec/img) as the cost indicator. As an example, comparing D48 (a ViT with 48 layers) to W3072, (a ViT with FFN dimension 3072 and QKV dimension 768 split across 8 heads) in terms of FLOPs or number of parameters, they have more or less similar cost, suggesting the D48 as a clear Pareto efficient model[2]. However, when looking at the speed (msec/img), we observe that W3072 is not necessarily worse than D48, since there are less sequential and more parallelizable operations.

**Target platform, and implementation** In some cases, a certain design in the hardware may lead to discrepancies between the cost indicators. As an example, tensor factorization is used as one of the common techniques to accelerate the matrix multiplication and used for making neural network models more efficient (Zhang et al., 2015b; Jaderberg et al., 2014). Weight decomposition, regardless of its effect on the quality of the model, can reduce the number of FLOPs in neural networks by 75% (Zhang et al., 2015a). However, it has been shown that using weight decomposition on GPUs can be slower as CUDNN that is optimized for $3 \times 3$ (He et al., 2017) convolutions and they prefer single large matrix multiplication instead of several small ones. FNet (Lee-Thorp et al., 2021) also shows how certain architectures can have significantly different speed in different hardware (e.g., GPUs and TPUs). When using a specific hardware and compiler, some small design choices can significantly affect the cost of a model. As an example, Zhai et al. (2021) show that for ViT, using global average pooling instead of a CLS as the representation of the input can significantly reduce the memory cost on TPU-V3. This is because the current TPU hardware pads the length dimension of the inputs to a multiple of 128, which may result in up to a 50% memory overhead. Another example for how implementation details can affect the efficiency is presented in (Vaswani et al., 2021), where the author discusses how carefully designed implementation of the local neighborhood gathering function for local attention might reduce memory usage while avoiding unnecessary extra computation.

---

[2]A model counts as a Pareto efficient model when having lowest cost compared to all the models with similar quality, or highest quality compared to all the models with similar cost.

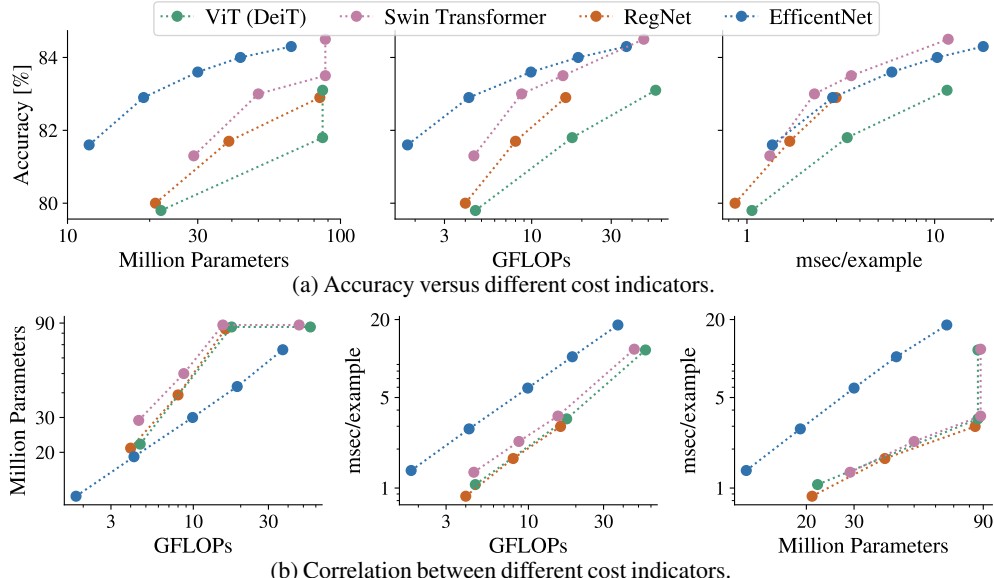

(a) Accuracy versus different cost indicators.

(b) Correlation between different cost indicators.

Figure 5: Accuracy and value of different cost indicators for different models on ImageNet dataset. Values for accuracies and cost indicators are obtained from (Steiner et al., 2021; Liu et al., 2021). Cost indicators are based on PyTorch implementation of the included models and the throughput is measured on a V100 GPU, using timm (Wightman, 2019).

## 3 DISCUSSION

The most common use of cost indicators in the ML community is for (1) comparing different models in terms of efficiency using a cost indicator, (2) evaluating different models in terms of quality while fixing the cost across models to have a fair comparison, and (3) choosing models with the right trade-off for the context at hand, e.g, in architecture search. While (1) and (2) are just two sides of the same coin, the focus in the first case is on finding models with better quality, while the second case is concerned with comparing quality while keeping a certain efficiency metric constant.

### 3.1 ON PARTIAL CONCLUSIONS FROM INCOMPLETE COMPARISONS
As we showed in Section 2, claiming a model is more efficient by reporting better scores on a subset of cost indicators can lead to partial, incomplete and possibly biased conclusions.

Figure 5a compares the FLOPs, parameters and run time for several models versus their accuracy on the image classification task. As seen previously in Figure 1, the relative positions of different model architectures are not consistent as the performance metric is varied. For example, on one hand, EfficientNets (Tan and Le, 2019) are on the Pareto-frontier of accuracy-parameter and accuracy-FLOPs trade-offs. On the other hand, the accuracy-throughput curve is dominated by SwinTransformers (Liu et al., 2021). As such, there is no model that is clearly more efficient here and conclusions may differ strongly depending on which efficiency metric is employed.

Note that when comparing variants of a single model family, most of cost indicators correlate. Hence, it could be sufficient to consider one of these cost indicators [3]. However, comparisons using a single cost indicator *between* models with completely different architectures are more complicated. Figure 5b shows the correlation between the cost indicators.

For example, we can see that for a similar number of FLOPs, EfficientNet has fewer parameters than other model families such as RegNet and SwinTransformer. Conversely, for a similar number of FLOPs, EfficientNet has slower throughput (i.e., higher msec/examples) than RegNet and SwinTransformer. These differences are not surprising, given that we are comparing transformer-based (Dosovitskiy et al., 2020; Liu et al., 2021) and convolution-based architectures (Tan and Le, 2019; Radosavovic et al., 2020). Moreover, some of these models are found via architecture search when optimizing for specific cost-indicators (e.g., EfficientNets are optimized for FLOPs).

Another observation from Figure 5 is that some variants of transformer based models have the same parameter count, while significantly different GFLOPs, throughput and accuracy. This is due to the

---

[3]Note that this is not always the case, as we have observed in Figure 4 that scaling depth vs depth of a single model can lead to different observations with respect to different cost indicators.

change in the model's input resolution (e.g., from $224 \times 224$ to $384 \times 384$), leading to different number of input tokens to the transformer encoder. This again indicates how a change in the setup could affect some of the cost indicators significantly while barely impacting others.

## 3.2 MAKING FAIR COMPARISONS THROUGH AN EFFICIENCY LENS

Efficiency metrics and cost indicators are often used to ground comparisons between two or more models or methods (e.g., a proposed model and baselines). By keeping one or more cost indicators fixed, one would often list models side by side in order to make comparisons fair. There are two main strategies here, namely (1) *parameter-matched* comparisons, where configuration of all models are chosen to have similar number of trainable parameters; and/or (2) *flop/compute matched* comparisons, where all models have similar computational budget, e.g., configurations are chosen to have similar FLOPs. Whilst there is no straightforward answer to which strategy is the better one, we highlight two case studies and discuss implications and general recommendations.

### 3.2.1 THE ISSUES WITH PARAMETER MATCHED COMPARISONS

We delve into some of the potential issues with the parameter-matched comparisons. The gist of many of these issues is that *not every parameter is created equal*, causing many intricate complexities that might complicate fair comparisons among models. Here we present situations where parameter matched comparison could go wrong.

**Token-Free Models** Token-free models (Xue et al., 2021; Tay et al., 2021c; Jaegle et al., 2021) get rid of the large subword vocabulary by modeling at the character or byte level. A large number of parameters originating from the embedding matrix is therefore dropped when transiting to token-free models. Hence, it remains an open question of how to fairly compare these class of models with their subword counterparts. ByT5 (Xue et al., 2021) proposed to up-scale the Transformer stack in order to compensate for lost parameters in the embedding layer. While this argument was made in the spirit of 'fairness' (e.g., comparing both methods at a parameter-matched setup), the up-scaling causes a *substantial* reduction in model speed. This is largely because parameters in the embedding matrix typically do not incur much computation while increased parameters in other parts of the Transformer stack (more depth/width) lead to a relatively substantial compute cost. What is referred to as *base* or *small* size here refers to a model that is significantly more computationally costly (in terms of speed, throughput and FLOPs) than their other counterparts and hence the naming alone can be very misleading to practitioners. To make this fair, ideally, the authors would have to present results that are both parameter matched *and* compute-matched .

**Encoder-Decoder vs Decoder-Only Transformers** When it comes to pre-trained language models, the choice of backbone architecture, encoder-decoder vs decoder-only, plays an important role in the efficiency of the model with respect to different cost indicators. When comparing these models, we need to know that an encoder-decoder model with $L$ encoder and $L$ decoder layers has approximately a similar amount of parameters as a decoder-only model with $2L$ layer, while it has half a compute and is twice faster (Raffel et al., 2019). This is due to a form of model sparsity in encoder-decoder architecture. Thus parameter-match comparison in this setup might be unfair to encoder-decoder models, especially when the speed is more of a concern than the memory consumption, e.g., when scaling up.

**Sparse Models and Mixture-of-Experts** A defining feature of sparse models (Lample et al., 2019; Fedus et al., 2021) is that they remain compute-matched while enabling scaling to a large number of parameters. Hence, parameter matched comparisons do not make sense for sparse models and parameter-matching sparse models can be seen as an unfair method of unnecessarily downplaying the strengths of sparse models. This has been also shown in Figure 1, where comparing a switch transformer with a vanilla transformer with the same number of parameters always goes in favor of vanilla transformer. Note that many works in the literature employ compute-matched comparisons for comparing sparse models (Narang et al., 2021; Fedus et al., 2021; Lample et al., 2019).

**Vision Transformers and Sequence Length** Models with a flexible sequence length, such as ViTs when varying patch size, have the opposite property of sparse models: one can instantiate architectures with significantly different computational cost (e.g., FLOPs and speed) while keeping the parameter count similar. Thus, this type of models should not be compared based on parameter count either. Table 1 presents the number of parameters, FLOPs, and inference speed of ViT using different patch sizes. We can see inverse an correlation between parameter size and both FLOPs and speed. Model

| Model | Input sequence length | Million parameters | GFLOPs | msec/example |
|---|---|---|---|---|
| ViT-B/8 | 785 | 86.5 | 78.54 | 7.17 |
| ViT-B/16 | 197 | 86.6 | 17.63 | 1.30 |
| ViT-B/32 | 50 | 88.2 | 4.42 | 0.39 |
| ViT-B/64 | 17 | 95.3 | 0.93 | 0.11 |

Table 1: Parameters size, FLOPs, and speed of ViT-Base with different patch sizes (i.e., $8 \times 8$, $16 \times 16$, $32 \times 32$, and $64 \times 64$). Models are fed by input images of size $224 \times 224 \times 3$, thus the input sequence length is $(224/\text{patch\_size})^2 + 1$ (for the CLS token). Numbers in the table are reported using the code in Scenic (Dehghani et al., 2021a) when running on 64 TPU-V3.

with bigger patch sizes have less FLOPs and are faster, while having more parameters, due to the larger patch embedding module.[4]

### 3.2.2 THE ISSUES WITH COMPUTE MATCHED COMPARISONS

We have previously discussed how parameter matched comparisons may be unfair. This section discusses a case where compute matched comparisons may raise concerns. Consider a scenario where the proposed method achieves compute saving by an architectural design that does not influence model parameters at all (e.g., downsampling sequences). A seemingly fair comparison here would be to take a standard model and remove layers or hidden dimensions until both models are compute matched. However, this comparison runs the risk of a baseline model that is handicapped by significantly insufficient model capacity in terms of parameter count and therefore, substantially underperform the proposed method. This is evident in Perceiver IO (Jaegle et al., 2021) where the baseline BERT is shrunk to a mere 20M parameters (compared to 425M in the proposed approach) in a compute matched comparison setup. Moreover, the baseline BERT is also handicapped in terms of depth (6 layers vs 40 layers) which is shown in (Tay et al., 2021b) to be unfavorable. Note that depth is not taken into account for FLOP-matched comparisons and if the authors were to account for speed-match, then the baseline here would be substantially faster than the proposed method. Overall, we do note that making fair compute matched comparisons is clearly nontrivial and a challenging problem. However, our recommendation is that when there is just no easy way to make a fair comparison, we encourage authors to make the best effort in finding a best *'matched'* setup and show multiple alternatives if possible.

### 3.3 COST INDICATORS FOR ARCHITECTURE SEARCH

Aside from being used for comparing models, many architecture search studies add a cost indicator to the loss function as a resource constraint (He et al., 2021b) and account for efficiency. To this end, these studies mostly use the parameter size (Pham et al., 2018), FLOPs (Hsu et al., 2018), memory access cost (MAC) (Ma et al., 2018), or real latency (Tan et al., 2019; Wu et al., 2019). Given that we have shown earlier how cost indicators may disagree with one another or lead to impartial conclusions, we suggest that practitioners place extra caution in choosing cost indicators for architecture search algorithms especially given its computational cost and sensitivity to the selected cost indicator. Here, making assumptions that cost indicators are always interchangeable runs the risk of conducting a massive search for finding models that are impractical.

## 4 SUGGESTIONS AND CONCLUSION

A lot of recent work has focused on comparing different model architectures on the basis of a cost indicator like parameter count or FLOPs. We have demonstrated in this paper that using any cost indicator alone can be misleading, with parameter count being the most problematic one. Oftentimes, parameter count is used to imply "model capacity"; however, when varying model architecture in any nontrivial way, this is wrong. Correctly estimating and comparing capacity across model architectures is an open research problem.

Since each indicator stands for something different and comes with its own pros and cons, we suggest always reporting *and plotting curves* using all available cost indicators, and refraining from highlighting results using just a single one. Moreover, given that it is basically impractical to provide a holistic report of all cost metrics (for instance, runtime on all possible hardwares) narrowing down the efficiency claims to the exact setup that models are evaluated and avoiding overgeneralized conclusions in comparisons can already provide a much more clear picture to the community.

---

[4]Using patch size of $n \times n$, the number of parameters of the patch embedding module is $n \times n \times 3 \times \text{emb\_dim}$, e.g., for ViT-B/64, it is $64 \times 64 \times 3 \times 768$.

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

# Appendix

## A  EXPERIMENTAL SETUP: SCALING DEPTH VS. SCALING WIDTH

Detailed configurations and scores for the experiments on scaling Vision Transformer, by increasing the depth vs increasing the width of the model.

| Model | Num layers | FFN dim | QKV dim | Num heads | Million Parameters | GFOPs | msec/img | ImageNet Accuracy |
|---|---|---|---|---|---|---|---|---|
| D6 | 6 | 1024 | 384 | 6 | 18.89 | 0.61 | 0.09 | 37.5 |
| D8 | 8 | 1024 | 384 | 6 | 22.44 | 0.79 | 0.11 | 42.4 |
| D16 | 16 | 1024 | 384 | 6 | 36.63 | 1.52 | 0.22 | 51.5 |
| D24 | 24 | 1024 | 384 | 6 | 50.83 | 2.25 | 0.32 | 55.7 |
| D32 | 32 | 1024 | 384 | 6 | 65.03 | 2.98 | 0.43 | 58.8 |
| D48 | 48 | 1024 | 384 | 6 | 93.42 | 4.43 | 0.64 | 61.8 |
| W768 | 12 | 768 | 192 | 3 | 9.47 | 0.31 | 0.11 | 34.4 |
| W1024 | 12 | 1024 | 384 | 6 | 24.81 | 0.92 | 0.16 | 45.2 |
| W1536 | 12 | 1536 | 512 | 8 | 42.51 | 1.70 | 0.22 | 50.3 |
| W3072 | 12 | 3072 | 768 | 12 | 101.52 | 4.44 | 0.35 | 58.3 |
| W4096 | 12 | 4096 | 1024 | 16 | 173.10 | 7.80 | 0.68 | 63.3 |

The above table has spanning headers: Configuration (Num layers, FFN dim, QKV dim, Num heads), Cost (Million Parameters, GFOPs, msec/img), and Quality (ImageNet Accuracy).

Table 2: Detailed configuration as well as cost versus quality scores for experiments on scaling depth or width of Vision Transformer.

## B  ADDITIONAL COST INDICATORS

Depending on the use case, there are other cost indicators that may play a key role in determining the efficiency of a model:

**Sample efficiency** that can be expressed as training data size, e.g. number of data points (or tokens for language models Kaplan et al. (2020)) that a model needs in order to reach a reasonable performance. Being sample efficient depends on many factors, like inductive biases of the model or the training curriculum.

**Carbon footprint** of a model, during training or inference that is a proxy of the environmental impact. Quantifying this cost directly is not straightforward, but (Strubell et al., 2019) proposed the approximate environmental costs of an ML model as $\text{Footprint} = (ee_{train} + \text{queries} \times ee_{inference}) \times CO2e_{datacenter}/\text{KWh}$, where, $ee$ indicate the electrical energy (Patterson et al., 2021).

**Monetary**, i.e., the expenses for training or serving models, which is reported in the form of figures (Sharir et al., 2020). As an example, the figures for training can be computed as total-train-time $\times$ total number of chips $\times$ prices offered by cloud solutions[5]. This cost indicator is more often used in business documents, than in scientific articles.

**number of model activations**, proposed by Dollár et al. (2021) as a complexity metric. Model activations refers to the number of elements in the output tensors from the building blocks of the model, e.g., output size of convolutional layers in ResNet. Dollár et al. (2021) argue that the number of activations is a reliable predictor for the model runtime and showed that compared to FLOPs, it is more strongly correlated with the runtime on memory-bandwidth limited hardware.

**Memory consumption** is one of the main dimensions of efficiency and it has been used as a cost indicator in various research works (Bondarenko et al., 2021; Dosovitskiy et al., 2020; Kondratyuk et al., 2021). Reporting memory footprint often is done in form of "peak memory usage", during training that takes into account the memory consumption by the model, optimizer, and the pipeline. Another form that is also common is comparing the maximum batch size that fits on a specific device for a specific task. Unlike model activation size, memory footprint depends on the hardware and implementation. It has a direct implication of whether a model can fit on a device, which is a hard constraint in some cases.

---

[5]Note that the prices largely depend on the price of electricity.

