# OpenReview forum: "The Efficiency Misnomer"
_ICLR.cc/2022/Conference — ICLR 2022 Poster_

### Official Review · Reviewer_sinW · 2021-10-29

**Correctness:** 3
**Technical Novelty And Significance:** 2
**Empirical Novelty And Significance:** 2
**Recommendation:** 5
**Confidence:** 4

**Main Review:**

The motivation behind the work is clear and valuable to the community. The authors provide relevant reflections and point out common mistakes. However, I believe that the technical content of the paper is not sufficient for acceptance at ICLR. While warning the community is surely important, I would encourage the authors to propose more  practical and detailed guidelines, hence, provide some complete and detailed examples of how model efficiency should be assessed and compared. It would be interesting if such examples were taken from current papers that draw wrong conclusions on model efficiency.

I would suggest to reduce the “Introduction” section (which contains come repetition) and  the “training or inference cost?” section (especially the "Gaming with training time" subsection) and enlarge the discussion section by giving complete and practical example of how model efficiency should be quantified and how two models should be properly compared. As already mentioned, it would be interesting if such examples were taken from current papers that draw wrong conclusions on model efficiency.

The authors should clearly define what “model capacity” is. They mention it, but do not define it properly.
The authors should clearly define “parameter-matched” and “compute-matched”.

In figures 1 and 2, apart from the parameters of the models that are changed, all the other parameters are set to the default values given by the referenced papers? It should Please specify if this is the case.

What is the hardware used for computing the msec/example in figures 1 and 2? Please specify it.

MINOR COMMENT:
Add a reference to Mesh Tensorflow.
Typo: Figure 3. “Ac curacy”; in Section 3.1 the figure number is missing in the text.


**Summary Of The Paper:**

The paper address the problem of model efficiency indicators. The paper brings up the risks of reporting only few efficiency indicators and points out this erroneous common practice.
The authors investigate how reporting only few cost indicators might lead i) to partial or incorrect conclusions about the model efficiency and  ii) to unfair model comparison. Finally, the authors give recommendations about how to report efficiency indicators.


**Summary Of The Review:**

I agree with the authors that it is important to raise awareness and to point out the issues related to the adoption of only a few cost indicators. I believe that the key message of the paper is valuable to the community. However, I believe that the paper does not qualify for acceptance at ICLR because of its limited technical contribution.

---

> ### Author Response · Authors · 2021-11-20
> **Response to Reviewer sinW**
>
> Thanks for the time you spent reading the paper and for the comments and suggestions. We have updated the paper with the following changes regarding your comments:
>
> * We have updated the paper, to incorporate suggestions and comments.
> * We added more examples and extra experiments for the arguments/discussions we have in the paper (e.g., Section 3.21).
> * We added details on the configurations we used for our experiments (e.g., Section 3.2)
> * We added details to make sure that the exact setup, hardware and software used for experiments in Figure 1 and 2 is clear to the readers.
> * We also provided definitions for “parameter-matched” and “compute-matched” in Section 3.2.
> * Rewrote some parts of the introduction to make the contributions more clear,  resolved typos, and added more details about the included examples/experiments.
>
> In response to "Reduce the introduction section and enlarge the discussion [...] It would be interesting if such examples were taken from current papers that draw wrong conclusions":
>
> Thank you. We believe that this is already addressed in Section 3.  For example, we discussed the efficiency of EfficientNet which is not necessarily the case when looking at the throughput as the cost indicator (Section 3.1), or ByT5 that uses parameter matched comparison while it is unfair as this led to allocating less compute to the baseline (Section 3.2), or Perceiver IO that uses compute matched comparison that led to shrinking the baseline in terms of parameter count, which is not fair either (Section 3.2).
> We grouped and organized more of these examples  in subsections of Section 3 and we tried to outline what went wrong in the comparisons and discussed the correct alternatives.
>
> We hope that the reviewer reconsiders the assessment based on the revised version of the paper with many improvements and are happy to incorporate more comments and suggestions.

---

> > ### Comment · Reviewer_sinW · 2021-11-24
> > **Answer to authors' reply**
> >
> > First, I would like to thank the authors for their answer and for incorporating the feedback.
> >
> > I believe that the new version is an improvement. However, I am still not fully convinced with regards to the novelty and contributions of this work as the authors present the issue of erroneous efficiency assessment and warn the community about this, but no practical and detailed guidelines to address this issue are proposed. Anyway, I have raised the score of my evaluation in reference to the improved revised paper.

---

> > > ### Author Response · Authors · 2021-11-30
> > > **Response to additional comments by reviewer sinW**
> > >
> > > Thank you for your considered response.
> > >
> > > We understand that the main suggestion is to include concrete guidelines for researchers when reporting efficiency metrics in their papers. In fact, we considered precisely this (along with a flowchart to follow) when originally writing the paper and spent quite a long time discussing this, as our ultimate goal is to help the community to make progress based on rigorous scientific works. However, this was not obvious as there are numerous variables with several dimensions (i.e., assessing efficiency on all possible hardware configurations). Furthermore, as detailed in Section 3.2, making fair comparisons to previous methods based on efficiency metrics is non-trivial and requires careful consideration by the researcher, and is often dependent on the model and problem domain. As a result, we refrained from providing a concrete guideline which may not cover all possible scenarios. Instead, we chose to provide several instances of “things that should not be done” in various common scenarios, tips on how to avoid these mistakes, and how to adjust claims to be correct, which we believe is a practical plan.

---

### Official Review · Reviewer_AyyT · 2021-10-31

**Correctness:** 4
**Technical Novelty And Significance:** 2
**Empirical Novelty And Significance:** 2
**Recommendation:** 6
**Confidence:** 4

**Details Of Ethics Concerns:**

No ethics concerns.

**Main Review:**

The main strength of the submission is that it clarifies and demonstrates the insufficiency of any single cost metric regarding neural network model efficiency. It lists a fairly broad set of cost indicators in Section 2, then focuses on demonstrating issues with the most common ones in the subsequent sections. Three sets of experiments (Fig. 1, 2, 3) show that different cost indicators may not necessarily be correlated.
The paper is overall clear and well written and lays out potential issues with evaluation and reporting of (and resulting claims from) the various indicators precisely.

While the insights provided in this submission may not be novel, this is the first paper to provide a comprehensive argumentation of advantages and disadvantages of cost indicators. There is little research insight beyond this, which, given the “survey-like” nature of the submission, is acceptable. The main weakness of the submission is the brevity of the discussion and suggestions (Section 4), which reduces to a plea to do rigorous science (report multiple metrics, don’t overstate claims).

Minor comments:
- On page 8, first paragraph, a figure reference is missing (presumably to Fig. 3).
- Pg. 9, last paragraph: wholistic -> holistic

**Summary Of The Paper:**

The paper provides a detailed discussion on various cost indicators for “efficiency” of neural network models. It looks at both training and inference processes and argues, based on several examples and arguments, that no single cost metric is sufficient to provide a complete picture of a models’s efficiency, due to the behavior of cost indicators not necessarily being correlated or highly dependent on the platform. It concludes with recommendations to include multiple cost indicators in papers and to clearly state (restrict) efficiency claims.

**Summary Of The Review:**

The submission contains an assessment of advantages and drawbacks of various model efficiency cost indicators. It comprehensively lays out issues and argues for a more rigorous and precise handling of reported metrics in academic work. Novelty and research insights remain limited, yet publication has good scientific value for means of awareness, reference and citation.

---

> ### Author Response · Authors · 2021-11-20
> **Response to Reviewer AyyT**
>
> Thanks for the time you spent reading the paper and for the comments.
>
> The main issue raised by the reviewer is the lack of novelty and the question if the paper is a good fit for an ML venue like ICLR. We have already discussed this in our general response. We would like to further mention that we agree that our study does not fall in the category of work presenting a  novel method, but given that “Novelty And Significance” is what is being evaluated, we believe our study brings enough value to be considered a significant work. We also believe that it can also serve as a self-contained reference that can be shared with new authors making the mistakes we outline.  As mentioned by one of the reviewers:
> > “the ML community is in dire need of work that explains relevant quantities and considerations, provides concrete examples of pitfalls and poor practices, and emphasizes the importance of context and nuance when assessing efficiency”.
>
> We believe ICLR is a great venue for such a study as it’s well respected by researchers and practitioners that would benefit from the discussions in our paper.
>
> We have fixed the minor comments and improved the paper and hope that the revised version of the paper and our response is considered by the reviewer for reevaluating the given scores. We are happy to also incorporate any concrete suggestions.

---

> > ### Comment · Reviewer_AyyT · 2021-11-21
> > **Thanks for the revision**
> >
> > I thank the authors for their continued work on the submission! While I took note of the limited novelty, it is actually not a major criticism in my review. In contrast, I find this "acceptable" and note "good scientific value for means of awareness, reference and citation".
> >
> > I found the main weakness to be the limited discussions of potential strategies to alleviate the problem. This discussion seems to not have been extended in the (fairly minor) revision; I appreciate that this is hard.
> >
> > Nevertheless, as before, I remain in favor of acceptance of this paper to ICLR, precisely because it is a venue where the presented insights can find the appropriate audience.

---

### Official Review · Reviewer_6vKw · 2021-11-02

**Correctness:** 3
**Technical Novelty And Significance:** 2
**Empirical Novelty And Significance:** 2
**Recommendation:** 5
**Confidence:** 4

**Main Review:**

Strengths:
-	Good overview (systematization) of efficiency metrics and the connection between them
-	Cases of misleading report in recent ML/AI studies

Weaknesses:
-	It is not clear the direct contribution to the AI/ML industry. The contribution looks as a meta-contribution to the environment in which the ML/AI science behaves. E.g., the metrics are known, the efficiency aspects to take care while considering ML/AI models are also well known. I believe any engineer from the industry takes care of any of the aspects. So, this paper states: “Please, report properly the result when publish”. For instance, in numerical math and algorithms, the differences between memory, parallelism, etc are well known and are not mixed.  For sure, there are works that report only few of aspects, but stating that they sometimes mislead, does not help the industry (this more relates to education?).
-	If this paper is about how the existing papers have misleading reporting, then it would be nice to have some quantitative analysis (based on some statistics). E.g., it is not clear how many times bad efficiency reporting occurs in the recent bunch of AI/ML papers.


**Summary Of The Paper:**

This work addresses the problem of measuring and reporting efficiency of machine learning models. The authors show that there is a series of efficiency metrics which are not fully correlated, but, in many of literature, only few of them are reported making confusing claims on production-level usage of proposed models. The main claim is a proposed list of recommendations for reporting of efficiency of new ML models and approaches.

**Summary Of The Review:**

Good paper that describes the issues in results reporting in ML/AI papers. However, given the weaknesses listed above, this work seems to have a notable room for improvement.

---

> ### Author Response · Authors · 2021-11-20
> **Response to Reviewer 6vKw**
>
> Thanks for the time you spent reading the paper and for the comments.
>
> We understand that the reviewer's primary concern is how this work would contribute to the ML industry (and broader community). We have already discussed this in our general response, while also nothing that the other reviewers endorsed the importance of such a work to the community:  Reviewer sinW: "I believe that the key message of the paper is valuable to the community"; Reviewer AyyT: "publication has good scientific value for means of awareness, reference and citation"; Reviewer GqNw:  "The paper is well-motivated and attempts to fulfill a critical need in the field".
>
> We strongly believe that ML researchers in academia would benefit from this study for improving the rigorousness of reports on efficiency in scientific papers. This leads to serving  ML practitioners, who are in charge of making decisions on what model to choose to be deployed in real-world products.
> So the impact of fixing issues we discuss easily goes beyond academic research and has a significant effect on the industry. This is demonstrated by the fact that there are startups and dedicated industry sections emerging with a focus on efficiency (e.g., MosaicML that is directly pointed out by reviewer GqNw).
>
> Besides, we understand that some discussions in our paper seem trivial, but as we can see, there are many instances of papers (we cover more than 15 examples of them in our paper) with incomplete reporting of efficiency metrics and casting partial conclusions, which is hard to keep track of. This, for us, was a clear sign of the need for a study that organizes all the information, raises the most common issues, and backs up the arguments with examples and controlled experiments along with suggestions and recommendations.
>
> We uploaded a revised version of the paper that is improved by including more supporting experiments and examples for the statements. We are happy to incorporate any additional concrete suggestions and we hope that the assessment by the reviewer will be reconsidered.

---

> > ### Comment · Reviewer_6vKw · 2021-11-26
> > **Thank you for the response**
> >
> > Thank you for the response.
> > I've read the improvements and appreciate the efforts made to improve the paper.
> > Meanwhile I would agree with the reviewer "sinW" that the paper has still minor contribution and, I believe, raise a known problem (as I said earier a practitioner usually knows that the lack of info in a paper should be a signal to verify the results on their own data). However, the paper would benefit significantly from some guidelines or suggestions on how to improve.
> > For instance, maybe some proposal of structure / scheme of reporting to resolve the issues would create value for authors that report their results?

---

> > > ### Author Response · Authors · 2021-11-30
> > > **Response to additional comments by reviewer 6vKw**
> > >
> > > We thank the reviewer for the additional comment and for the discussion.
> > >
> > > We agree with the reviewer that the existence of such guidelines can be helpful. We described the difficulty of having such guidelines in [our response to reviewer sinW](https://openreview.net/forum?id=iulEMLYh1uR&noteId=2W9DubYl98K) and the reasoning behind our choices for the structure of the paper.

---

### Official Review · Reviewer_GqNw · 2021-11-03

**Correctness:** 4
**Technical Novelty And Significance:** 3
**Empirical Novelty And Significance:** 3
**Recommendation:** 8
**Confidence:** 4

**Main Review:**

The motivation for this paper is excellent. The measurement and discussion of “efficiency” in contemporary ML research is typically simplistic, and sometimes misleading. The ML community is in dire need of work that explains relevant quantities and considerations, provides concrete examples of pitfalls and poor practices, and emphasizes the importance of context and nuance when assessing “efficiency”. Detractors might accuse this paper of not containing sufficient original research for publication in a venue such as this one. My response to this criticism is that ML research would be in a better place if researchers took more time to reflect on the state of research and identify ways in which our practices and scholarship could be made more robust and rigorous. That is what this paper does.

Unfortunately, I don’t think this paper fully realizes its potential. It could be clearer and more focused; it reads like it was written quickly. Many of the examples don’t feel sufficiently explained, and some don’t feel well-integrated. In its current state, I am not confident that it is suitable for publication. However, I am optimistic that it could be after revision.


**Feedback**

My primary criticism is that the paper is insufficiently concrete. It would be substantially more impactful with more real examples, ideally with accompanying plots and/or tables. For example, on page 5 the authors state

>Some hyper-parameters, such as learning-rate and weight-decay, can be tuned so as to reach good quality quickly but then plateau lower than in “slower” settings that eventually reach higher quality...it could well be that method B gets to X+epsilon faster than A, or worse, A may never reach X+epsilon.

These concepts could be illustrated with a plot showing learning curves for different hyperparameters and/or methods, ideally with real data. This also applies to the case studies presented in Section 3.2.1, all of which currently feel too light, and would be greatly improved with plots and/or tables.

The ViT model configurations section *Degree of parallelism: Scaling Depth (D) vs. Scaling Width (W)* requires additional explanation. Do all of the other hyperparameters remain constant? What is the width for the variable-depth models, and what is the depth for the variable-width models? Furthermore, the authors show a wide range of depth and width values but only discuss one depth-width pair. I think a more accurate and meaningful statement is something to the effect of “When using FLOPs or # params as a metric, increasing width is more beneficial for smaller param/FLOPs budgets, while increasing depth becomes more beneficial for larger param/FLOPs budgets. However, this effect is much less clear when using msec/image as a metric. Performance is similar for when scaling width vs. depth”. Additionally, the effects of scaling width vs. depth could become clearer if the authors sampled more datapoints between D6-D32, and beyond W3072.

At the top of page 8, the authors state “for a similar number of FLOPs, EfficientNet has fewer parameters than RegNet and SwinTransformers.” This statement would be more impactful if FLOPs and # params were directly plotted against each other.

There is at least one company—[MosaicML](www.mosaicml.com)—founded with the goal of improving the efficiency of training deep learning models. Their research seems relevant—they present a [framework for analyzing “efficiency"](https://www.mosaicml.com/blog/methodology). The existence of this company could be taken as evidence in favor of the relevance of the present work, and as such the authors may want to discuss/cite it.

Page 6, *Introducing Sparsity*. The authors write

>...while for unstructured sparsity, it is not possible for the corresponding low-level operations to reach the same efficiency as their dense counterparts on current hardware, where memory access is significantly more expensive than compute.

I think this point could be made more strongly: no common accelerator can extract significant efficiency gains from unstructured sparsity. This could be illustrated with real data showing the change in wall clock time per step as a function of % sparsity for two models, one in which the sparsity is structured and one in which the sparsity is unstructured, ideally for multiple different accelerators.

On page 5, the authors state

>However, counting training cost in terms of steps can be problematic as step count in itself is not a meaningful metric, and can be stretched arbitrarily with optimizers introducing multi-step lookahead schemes.

This statement could also be emphasized or strengthened. I’m having a hard time thinking of a scenario in which step count is informative.

The statement of contributions (2nd to last paragraph in intro) should be more specific. I suggest the authors enumerate (or provide examples of) specific observations and suggestions/recommendations.

The final paragraph of the introduction feels redundant.

The concept of “Pareto-optimality” is used but not explained. This concept should be explained if the present work is attempting to be a go-to reference for understanding efficiency.

The authors detail many different efficiency metrics, which is excellent. However to make room for the additional text and figures that I think this paper needs, I suggest moving to the appendix descriptions of metrics that aren’t used elsewhere in the main text. I also think Section 3.3 (Cost Indicators for Architecture Search) could be expanded and moved to the appendix.

Page 6, “Sharing parameters”: The authors write “When we introduce a form of parameter sharing, it is clear that compared to the same model with no parameter tying...”. Please be consistent in your terminology; pick “parameter sharing” or “parameter tying” and use one exclusively.

Typo: “These differences are not surprising, given that Fig. is...” the Figure is not referenced. I assume it should “Fig. 3”.


**Summary Of The Paper:**

This paper emphasizes the importance of context and nuance when discussing “efficiency”. The authors explain different efficiency metrics and the distinction between training and inference efficiency. They then illustrate how assessments of the “efficiency” of different models can be misleading or even contradictory depending on factors such as the choice of efficiency metric, baseline, architecture, hardware, optimizer, and other experimental design choices. The authors finish with a few suggestions for best practices.

**Summary Of The Review:**

The paper is well-motivated and attempts to fulfill a critical need in the field, but feels a bit rushed and incomplete in its current form.

---

> ### Author Response · Authors · 2021-11-20
> **Response to Reviewer GqNw**
>
> Thank you very much for your thoughtful comments and the substantive review. We appreciate that you find our paper impactful and glad to see your argument on why ML venues are good home for such a study. This is indeed motivating that the value of our work is recognized and we really hope to see an agreement on this among others.
>
> We went through your comments, one by one, and revised the paper with the following changes:
> * We added a concrete example  on how  the learning rate decay under shorter vs longer schedules lead to faster or slower convergence, while having impact on the final performance (Figure 2). We also added an empirical example  on how the choice of hyper parameter may lead to slower but better convergence, versus faster, but worse convergence (Figure 3). Both are  under “Gaming with training time”.
> * Added an experiment/example to support the argument on “Vision Transformers and Sequence Length”  in Section 3.2.1: Table that presents different cost metrics for ViT-B/8, ViT-B/16, ViT-B/32, and ViT-B/64.
> * Added three new experiments with the requested configs for “Degree of parallelism: Scaling Depth (D) vs. Scaling Width (W)”. The new configs are D8, D24, and W4096. Figure 2 is updated. Also the suggestion on providing more general observations from the figure is incorporated in the text.
> * Added reference to the example presented in Figure 1 for “Sparse Models and Mixture-of-Experts” in Section 3.2.1 which also supports the argument against parameter-match comparisons with sparse models.
> * Added a Table with detailed configurations, values of cost indicators, and performance score of all the models from the “Degree of parallelism: Scaling Depth (D) vs. Scaling Width (W)” section.
> * Added three new plots that show the correlation between different cost indicators to make the discussion in the Section 3.1 (“On partial conclusions from incomplete comparisons”) more clear and immediately observable.
> The introduction is updated with a more concrete list of contributions. We also removed redundant content in the last paragraph.
> * Provide definition for Pareto-optimality, fixed typos and missing references, resolved inconsistencies (e.g., sharing/tying),  and rewrote some small parts to improve the clarity.
> * Refactored the paper by moving some contents that are not core information to the appendix.
>
> Regarding your comment on comparing the effect of unstructured sparsity on various accelerators, there is indeed a great deal of values for such a study, and we believe it deserves a dedicated rigorous work, trying out various optimized libraries and carefully ablating varying amount of sparsity on different  accelerators, that goes beyond what we were able to cover in our paper.  For more context on this,  we referred our readers to the [great paper from Gale et al](https://arxiv.org/abs/1902.09574) which is all about implementing as efficient as possible unstructured-sparse kernels on hardware accelerators.
>
> Once again, thanks for the comments. We enjoyed reading your suggestions and sincerely hope that we have addressed your major concerns in the rebuttal. We would be happy to discuss if there are any further points. Considering that you find the impact of our paper significant, we appreciate it a lot if you can update your overall assessment score of the paper if there are no additional issues.

---

> > ### Comment · Reviewer_GqNw · 2021-11-22
> > **Updates**
> >
> > I think the updates to the paper really help to clarify the concepts and contributions. Thanks for all your work. I have updated my score from a 5 to an 8 accordingly.
> >
> > I still have a few minor comments:
> >
> > Section 3.2.1, first paragraph:
> > >Here we present two example situations where parameter matched comparison could go wrong.
> >
> > Looks like there are three examples.
> >
> > Why does increasing the patch size if a ViT from 8x8 to 64x64 increase the parameter count by ~10m? Am I forgetting something other than the larger patch embedding parameters? A brief explanation in the text could be helpful.
> >
> > In Figure 5, ViT and SWIN both have a large change in accuracy, GFLOPs, and msec/example despite a (relatively) fixed parameter count. This is visible in Figure 5a, left panel, and 5b, left and right panels. Perhaps I missed something in the text, but it would helpful to explain this.

---

> > > ### Author Response · Authors · 2021-11-23
> > > **Thanks for additional comments**
> > >
> > > Thank you for your additional comments. Also we appreciate you updated the scores based on the updates and responses.This is really motivating.
> > >
> > > We have uploaded an updated version of the paper based on your comments and suggestions. Also here are quick responses to your questions:
> > >
> > > > Looks like there are three examples.
> > >
> > > Thank you for pointing this out. We updated the paper to fix this.
> > >
> > > > Why does increasing the patch size if a ViT from 8x8 to 64x64 increase the parameter count by ~10m? Am I forgetting something other than the larger patch embedding parameters? A brief explanation in the text could be helpful
> > >
> > > You are correct. The difference mainly comes from the patch embedding parameters. Patch embedding is in fact a convolution with kernel size (and stride) equal to the patch size  we select. With ViT Base, the dimension of the embedding is $768$ (based on the reference paper).   When the patch size is $8 \times 8$, the number of parameters of the patch embedding module is $8 \times 8 \times 3 \times 768 = 147.5 \times 10^3$, while with the patch size of $64 \times 64$, it is $64 \times 64 \times 3 \times 768 = 9.44 \times 10^6$ (an increase by a factor of $64$)). We have added this to the paper to explain the difference between the parameter counts.
> > >
> > > > In Figure 5, ViT and SWIN both have a large change in accuracy, GFLOPs, and msec/example despite a (relatively) fixed parameter count. This is visible in Figure 5a, left panel, and 5b, left and right panels. Perhaps I missed something in the text, but it would helpful to explain this.
> > >
> > > Thank you for the suggestion. We updated the text in Section 3.1 to discuss this. It is actually due to increasing the input resolution of the model (from $224  \times 224$ to $384  \times 384$). This barely changes the number of parameters (the only additional parameters are positional embeddings), but has a marked impact on FLOPs (increase), throughput  (decrease) and accuracy (increase).

---

> > > > ### Comment · Reviewer_GqNw · 2021-11-23
> > > > **Thanks for the clarification**
> > > >
> > > > Thanks for the quick response! The explanation of the effect of changing the input resolution aligns especially well with the overall message of the paper.

---

### Author Response · Authors · 2021-11-20
**General Response**

We would like to thank the reviewers for the time they spent giving us feedback.

We want to highlight that the take-home messages presented in this paper are of key importance for assuring healthy reporting of efficiency in machine learning research, and to the industry as a whole as machine learning becomes more and more productionized. Many of them may seem trivial after reading them structured and backed with examples, however, we are observing the issues that we discussed in this paper almost every day in the newly released papers with poor practices on analyzing efficiency. We strongly believe a study like this would help everyone to have a better understanding of the comparisons, both those who are presenting new ideas (researchers) and those who choose ideas to operationalize them (practitioners). It can also serve as a reference to educate any future authors making the mentioned mistakes, since such a reference aimed at an ML audience does not currently exist.

We appreciate the statement from review GqNw on this:
>“Detractors might accuse this paper of not containing sufficient original research for publication in a venue such as this one. My response to this criticism is that ML research would be in a better place if researchers took more time to reflect on the state of research and identify ways in which our practices and scholarship could be made more robust and rigorous. That is what this paper does.”

Besides, we emphasize that this paper is the outcome of numerous discussions with various researchers, many observations collected over time while experimenting and working on new ideas both in vision and language, reading through the literature, and comparing many works to understand what are the current issues. Time spent on this paper goes beyond the authors' list and we are happy that we were able to organize all this information, bring examples, run experiments to back them up, and present them to the community. The key objective for us is to see the impact of this study in future research.

We have revised the paper based on comments from reviewers, added more concrete examples and experiments to support the various arguments in the paper, and finally, here we respond  to individual reviewers' comments and questions.

---

### Decision · Program_Chairs · 2022-01-20

**Decision:**

Accept (Poster)

**Comment:**

Overall the reviewers like the ideas in this paper.  It calls out some of the issues with the current line of thinking in the ML/AI community.  There were some concerns, but overall this paper offers a new way to think about, present, and question efficiency results.  This could be quite infulential.  I think this is interesting enough to warrent publicaiton.